# New Conceptual Model of Social Sustainability: Review from Past Concepts and Ideas

**DOI:** 10.3390/ijerph20075350

**Published:** 2023-03-31

**Authors:** Anh M. Ly, Michael R. Cope

**Affiliations:** Department of Sociology, Brigham Young University, Provo, UT 84602, USA

**Keywords:** social sustainability, risk, social capital, resilience to climate change, adaptation

## Abstract

The social dimension of sustainability has remained relatively underdefined, despite the efforts to specify and integrate this dimension into the general sustainability conversation of scholars and practitioners. This study aims to advance the conversation of social sustainability by examining past the multi-disciplinary literature and policy documents, as well as proposing a comprehensive conceptual model of social sustainability. We present a model with five dimensions: safety and security, equity, adaptability, social inclusion and cohesion, and quality of life. Through these dimensions, we propose social sustainability as a process that strives for effective management and allocation of social capital as a constitutive resource, and the confrontation of such controllable and uncontrollable risks as natural disasters and climate change. Our model was constructed with the purpose of providing scholars, policymakers, and practitioners with a comprehensive guideline to create social sustainability policy with human beings as the priority and cultural awareness as a grounding approach to initiating disaster-related and climate-change resilience.

## 1. Introduction

With the introduction of a social dimension into the conversation of sustainability, multi-disciplinary scholars and practitioners over the past few decades have created a dynamic discussion of what a sustainable society should encompass, including urban sustainable development, corporate social responsibility, sustainable management, and weak/strong sustainability. There is active discussion within and between different disciplines, each claiming their significance in assessing social sustainability; however, there is much work left to be conducted [1,2]. In this study, our purpose is to provide a synthesis of considerations when attempting to create a conceptual model according to various contexts, cultures, and geographical factors. This study in particular highlights the congruency between a socially sustainable society and a resilient society, elaborating on adaption to climate-related natural disasters as one of the core elements in constituting a sustainable society. In doing so, it was helpful to trace the origin of the sustainability concept, capture its birth, and review its past definitions and leading discussions.

### 1.1. Historical Milestones of Developing the “Sustainability” Concept

Human history has suggested that the welfare of human beings has been a constant universal concern, one that has been recognized and built on from the extrapolations of present patterns and experiences from past civilizations. As human civilization has evolved, human progress and modernity have been perceived as linear, in which science is the ultimate tool for humankind to gain mastery over nature and grow stronger [3,4]. The rise of industrial capitalism in the Western world, which then spread to the rest of the world, has tied human progress to economic growth and material prosperity based on the advancement of science and technology. The transition toward a capitalist society remains a pessimistic viewpoint. Inequality between genders and among ethnicities, the uneven distribution of wealth, and environmental degradation have become threatening problems. Such progress has led to the rising concern of societal sustainability.

The concept of “sustainability”, although not necessarily coined as such, has been around for several centuries. Starting from the end of the 16th century until the beginning of 19th century, “Holznot”, which is roughly translated as “wood crisis”, occurred across central Europe, especially among German-speaking regions. In 1713, Hans Carl von Carlowitz discussed a type of forestry management that he called Sylvicultura Oeconomica (the Instructions for Wild Tree Cultivation). The shortage of wood as the main substance for fuel energy, machinery, and construction during that time urged authorities and researchers to take action, and the use of the term “nachhaltigkeit” (sustainability) was initiated for the first time by Carlowitz. The concept was described as a resolution in terms of forestry management with an ultimate goal of achieving the greatest possible efficiency of wood harvesting over time without overexploiting in the short term [5]. During the time of the Industrial Revolution, the first warning of environmental degradation was raised by George Perkins Marsh, an American philologist and pioneer environmentalist, in his writing of *Man and Nature*; or *Physical Geography as Modified by Human Action*. Specifically, the work addressed the desolation of nature resulting from a course of human action over history that could lead to an Earth unfit for human habitation [6]. With the slow and deep-seated transformation of an agricultural society into an industrial one, Marsh [6] argued for the importance of conserving nature as well as for a rational and mindful use of resources without abusing the Earth.

Along with the debut of the words “sustainability” and “sustainable” in the Oxford English Dictionary, the second half of 20th century witnessed a dynamic development of the sustainability concept. Awareness of the deadly Dichlorodiphenyltrichloroethane (DDT) by an American biologist Rachel Carson [7] in her famous book, *Silent Spring*, significantly marked the rising concern of environmental toxicity resulting from the massive use of pesticides in agricultural practices. Human poisoning, cancer, and illnesses, as well as the destruction of natural resources as the profound consequences of exposure to DDT pesticides, swayed public opinion and urged governments to take environmental concerns seriously and initiate environmental policies. *Silent Spring* created a significant environmental movement in policy-making discussions and raised public awareness to minimize the deadly effects of pesticides on human beings and ecosystems.

With debates in several industrial fields calling for future resolutions to the utilization of resources, the 1983 United Nations (UN) Commission on Environment and Development established *Our Common Future* (known as the Brundtland report), drawing attention to the definition of sustainable development. Specifically, the report marked the first introduction of the normative concept of “sustainability” into mainstream policy discussion by explaining it as the development that “meets the needs of the present without compromising the ability of future generations to meet their own needs” [8]. Even though some contemporary social issues were discussed in the report, such as world poverty and social inequalities and inequities, no specific definition of social sustainable development was proposed. Instead, the report explicitly explained the human utilization of resources and common concerns toward global ecology (environmental aspect) and economy (economic aspect). Based on the strategic imperatives from the Commission, the report suggested international, national, regional, and local spectrums of sustainable development, offering authorities and policymakers suggestions on how to acknowledge and achieve specific targets for sustainable development. The terminologies of “sustainability” and “sustainable development” are commonly used synonymously. Scholars have distinguished between the two concepts: “sustainability” is emphasized as a philosophy of long-term goals while “sustainable development” is defined as the “many processes and pathways” to achieve sustainability [9]. As a consequence, the policy-related discourse of the Brundtland report has played a part in the distinct utilization of the “sustainability” concept among scholars and the utilization of “sustainable development” concept among policymakers and corporate setups. Focusing on the social aspects of the two concepts, there is no evidence that academia and the policymakers are speaking the same language when it comes to what “social sustainability” or “social sustainable development” are. This is not to discredit either the policy movements or academic movements since both considered social elements in their understanding of sustainability concept. Instead, in following sections, we aim to identify how practitioners, policymakers, and scholars view social sustainability differently in the two following sections.

#### 1.1.1. Policy Movements

Governments and policymakers addressing social sustainable development specifically start and progress with diverse sets of goals, including targets and indicators to explicitly undertake contemporary common issues and concerns. In 2000, with the comprehensive concept of “sustainable development” in the Brundtland report as a baseline, the UN established the Millennium Development Goals 2000–2015 (MDGs). With the institution of eight comprehensive goals, in terms of social sustainability, the agenda aimed to address global poverty and hunger, provide primary education for all, promote gender equality and the empowerment of women, improve maternal and children’s health, and develop global partnerships [10].

This global agenda, in which social concerns were tackled separately, has been significantly impactful to governments and policymakers. For example, centering on community in social sustainable growth, in 2003 the office of the United Kingdom (UK) Deputy Prime Minister established an action program called *Sustainable Communities: Building for Future.* Even though sustainable communities, as defined by the program, are the relation between physical environmental settings and intangible values and psychological mores, the program mostly proposed a detailed agenda and action plan for improving only physical environments. Particularly, it addressed housing demand and supply, land utilization, and countryside and rural communities as the main themes in its action program, with timelines and resolutions for each challenge [11]. The intangible aspects of a community, which are believed to be essential for a sustainable society, remained unaddressed.

This pattern of addressing social sustainability specifically toward detailed physical goals has continued among policymakers, governments, and corporate settings. In 2015, with the completion of the MDGs, the UN established a 2030 global agenda with seventeen Sustainable Development Goals (SDGs), addressing diverse social problems. These issues are poverty and hunger (SDG.1 and SDG.2), health and well-being (SDG.3 and SDG.6), education (SDG.4), equality (SDG.5 and SDG.10), cities and communities (SDG.11), justice and peace (SDG.16), and global partnerships (SDG.11) [12]. Although the specification of sustainable goals is vital in the policy documents to provide orientations in several scales, the comprehensive understanding of the social sustainability concept is urgently pivotal for scholars and researchers, authorities and governments, and citizens to progress toward sustainable development unitedly and effectively.

#### 1.1.2. Academia Movements

While the discussion of sustainability with a specific focus on diverse social expressions has flourished among practitioners, academia seems to be less involved in the discussion of social sustainability. Social sustainability is a concept that requires efforts and research from multiple disciplines, especially the social sciences [13,14,15]. So far, economists (e.g., [16,17]), urban planners, and community developers (e.g., [18,19]) have been the most active, while sociologists, demographers, political scientists, and anthropologists should be heard from more frequently. The study of the idea has been scattered and diverse, creating a dynamic academic discussion and adding to the vagueness of the social dimension of sustainability [14,19,20]. Table 1 offers a number of representative movements of social sustainability, drawn from multiple disciplines and chronologically presented.

### 1.2. An Idea of Social Sustainability—The Absence of Mutual Understanding

To exclusively treat sustainability as only a matter of environmental degradation or conservation, economic growth and distribution, or a quantitative scientific concern is to “over rationalize both the problem and the solution” [15]. Thus, a healthy society is a fundamental condition for the economy and the ecosystem to thrive. The science of sustainability needs serious consideration of the human world, including human well-being, social values, notions of equity and happiness, social capital, and prejudices [15,25]. Therefore, the first objective of this study is to synthesize the existing literature and governmental documents relating to the social dimension of sustainability, extracting the most common elements discussed in both practical agendas and academic research works.

The second objective that we aim for in this study is to reconcile as well as advance the conversation of what a sustainable society should be. Boström [26] justified that the social pillar of sustainability is commonly seen as “a missing pillar”, indicating the dearth of academic discussion and the challenges to analyze, comprehend, and define the concept. Thus, as Table 1 exemplified oppositely, social sustainability appears to be diversely comprehended rather than to lack understanding. The pillar is itself a complex network of individuals, interests and values, and unfixed causes and effects, resulting in various implications on human life [20,26]. Additionally, as the historical timeline suggests, originating from global and political capitalism and rooting in environmentalism has marginalized the social dimension in almost every sustainability discussion [14,20,27]. Recognizing the current chaos among scholars, a comprehensive guideline addressing social issues in achieving sustainability is essential [19,22,27,28,29].

In this study, a comprehensive conceptual model of social sustainability will be presented and followed by a detailed justification of each element within the model. Each dimension is justified by general meanings, how risks, especially climate-related risks, perform within the element, and how social capital contributes to lower risks and enhances a sustainable society. We then discuss the characteristics of the model and prospective implications for scholars, governmental bodies, policymakers, and practitioners.

## 2. Materials and Methods

Given the emerging issues in conceptualizing social sustainability, we conducted this study using discussion drawn from such multi-disciplinary literatures as urban planning and community development, sociology, public policy, economics, business management, and ecology. Using “sustainability”, “sustainable development”, “social sustainability”, and “sustainable society” as the key search words, we narrowed our data resources to academic books, journal articles, and governmental policy publications, using online search engines (Google Scholars, Google Books, JSTOR, and PubMed). Research publications which are moderately impactful (mainly more than 100 citations) are sampled for our review study.

We employed grounded theory and qualitative coding techniques for data analysis. To analyze the data, we employed a deductive analytical strategy to construct our preliminary coding list. Specifically, safety and security, equality and equity, inclusion and coherence, and human well-being were key themes in the core coding list after skimming through the sampled literature. For detailed analysis, we conducted an inductive analysis method to openly seek for and identify trends and emerging patterns across the literature. After the literature were thematically sorted, we deep-read through each work of literature individually and coded them by hand according to the preliminary coding list. Employing open coding techniques [30], we also openly and flexibly generated new codes when applicable and compatible with the research themes. With non-existing and emerging themes that did not match, we deconstructed, categorized, and named new categories to accommodate the variation using selective coding technique [30]. For example, adaptability, general life satisfaction are the emerging concepts in the literature. The first draft of the conceptual model was then established after the first round of coding with the grounding elements of social sustainability extracted from the literature. We then extensively reviewed the materials for the second round and modified our conceptual model. After finishing with the coding process, we conducted a final review to extract the main description of each element that we would present in the study, using grounded theory technique with “constant comparison” and “densifying the theory” [30]. Each concept presented in the conceptual model is discussed dominantly as key dimensions of social sustainability in sampled works. Thus, each concept possessed its own attributes, assumptions, and limitations, capturing distinct perspectives as well as serving in specific functions in the final model.

While we do not claim to have collected every work of literature discussing social sustainability, we assert that the conceptual model below has crystalized the most significant studies examining various definitions, perspectives, and public policies of social sustainability. Accordingly, this paper is not merely a theoretical summary of social sustainability but a consistent model in which all elements are integral and have inter-correlations to construct a notion of social sustainability.

## 3. Results

### 3.1. Risks and Social Capital—Constitutive Elements of Social Sustainability

With the goal to conceptualize social sustainability, Eizenberg and Jabareen [2] emphasized risk as “the ontological foundation” of the idea, following a key thesis of Beck and Giddens of modern society as a “risk society” that is “a systematic way of dealing with hazards and insecurities induced and introduced by modernization itself” [31].

As a function of risk, modernity is argued to be shaped only threats and risks which might, in terms of the social, structural, and physical, cause harmful consequences to human society and its living environs. Thus, Beck and Cross [32], in their book, *Power in the Global Age*, characterized a risk society in modernity with three key dimensions: “(1) spatial, as reflected in the fact that many new risks (such as climate change) do not recognize the borders of nation-states and other such entities; (2) temporal, as manifested in the long latency period that is characteristic of new risks (such as nuclear waste), making it impossible to effectively determine and limit their effects over time; and (3) social, as exhibited in the complexity of the problems and the length of the chains of effect, which means that it is no longer possible to determine causes and consequences with any degree of reliability (as in the case of financial crises)” [2].

Giddens [33], aligning with the idea of seeing modernity as a risk society, identified two main types of risks: external and manufactured. Specifically, external risks are produced by a non-human source and normally are beyond human control. This category is identified with its nature of unexpectedness, regular frequency, massive scope and, to some extent, predictability. Natural disasters are one typical example of external risks, including devastating earthquakes, tsunamis, and volcanic eruptions. On the other hand, manufactured risks are defined by an elevated level of human agency, resulting as consequences of the modernization process itself. As a result of “intensifying globalization” [33], human beings have very limited or a total lack of knowledge and historical experience to confront and respond to manufactured risks [2], which are not limited to the social or economic dimensions of modern societies, but also pose negative impacts on ecosystems and climate change, and, in return, human lives.

Both Beck and Giddens were aware of both non-human risks and human-made risks. If viewing them from the aspect of human control, these two types of risks can be categorized as uncontrollable and controllable. Thus, Giddens [34], in his *Reith Lecture Series*, indicates that: “At a certain point, however—very recently in historical terms—we started worrying less about what nature can do to us, and more about what we have done to nature”. From Giddens’s idea, it is not hard to see the dominant spark of the idea of human impacts on both nature and their own living conditions; thus, human beings are the active agents of a society in which human beings build, run, and navigate the whole system [35]. In other words, a sustainable society can only be realistic if human beings are able to control what they can control effectively, such as social and economic assets and the consumption of natural resources through policy and legislation systems, as well as respond well to what they cannot control, such as natural disasters. While acknowledging the rising frequency of risks perpetuating within modern societies, we argue that considering risk as the only constitutive element of social sustainability is not enough to optimize and conceptualize the idea. Thus, in this study, we adopted the notion of sustainability and social capital theory to construct a more comprehensive foundation of social sustainability.

Tracing back to the history of the sustainability concept, it is not hard to spot several such typical phrases or terms as “shortage”, “overexploited”, “conserving nature”, and “meet the needs”. In the Brundtland report, a call for action emphasized that “the time has come to take the decisions needed to secure the resources to sustain this and coming generations” through which stating the importance of moderate consumption of and strategically allocated resources of the planet and human beings in every sense [8]. Scholars studying the idea of sustainability have indeed acknowledged the impossibility of infinite sustainable growth as well as the notion of moderation in every sector of a society on a finite planet [36,37,38]. Additionally, social capital, as we discuss in detail below with three interrelated forms, “is congruent” with the paradigm of sustainable development and the logic of cohesion [39]. The complex idea of capitals would “provide a framework within which to categorize, measure, and assess community and social change” [40]. For example, Cocklin et al. [40] broke down “capital” into key subsets for assessing the level of community sustainability in their research of six central rural areas in Australia. McKenzie [22] highlighted sustainability as an asset in itself, adopting from several researches as “occurring naturally and to varying degrees within societies, which allows them to maintain coherence and overcome change and hardship”. Following the virtue of sustainability, it is essential to conceptualize social sustainability with resources as the foundation. As social capital is often referred to be “the unrecognized development asset” [39], we suggest that to make a society sustainable, the management of social capital as the central resource and as an asset is vital.

Social capital, adopting from Bourdieu [41], Coleman [42], and Putnam et al. [25], is a collective resource “residing in the social structure of relationships among people” within a given society [43]. In our analysis for each author, the idea of social capital refers to different levels, including bonding, bridging, and linking levels. Firstly, Pierre Bourdieu’s idea of social capital lies in the benefits of individuals from participation in a social group or network, i.e., from attaining “membership in a group which provides each of its members with the backing of the collectivity-owned capital” [41]. His units of analysis are among individuals and families. Thus, social capital from Bourdieu’s viewpoint can be seen as a property of individuals from social networks and social group membership that are the resource facilitators for individuals’ advancement and development. Aligning with the idea of social sustainability, individuals are the agents of action as well as the essence of a society, eventually. In this sense, a sustainable society is desired to be a place where human beings can benefit in every aspect of life [18,19,24], allowing them to advance themselves using the social capital they possess. This is not to say that overexploitation and extra greed should be socially accepted; rather, it is about the improvement and advancement of human lives to a moderate level deriving from social capital. Social capital, in this level, is referred to as the “bonding” form, which is constituted from “the creation of informal associational networks, such as the extended family, neighbors, or cultural minority groups” [39]. This form of social capital is the most direct form of support and “solidarity-driven empathy and behavior” [39].

Secondly, for James Coleman, social capital is not the only type of private goods that only individuals can benefit from but is also a public good that benefits other individuals within a social group and a given community as a whole. In his work examining social capital in the creation of human capital, Coleman [42] explained how it is embedded with social context and structures (closure of social networks and social organizations) by which certain characteristics of social relations are facilitators of its appearance, including three main forms: (1) trust and reciprocity among members of a given inner group, (2) information channels, and (3) effective normative regulations. As a public good, social capital presents in the sense that the direct contribution of individuals can benefit a whole community. Empirical evidence suggests the benefits of community by fostering social capital through diverse communal activities align with Coleman’s justification of social capital (e.g., [44,45]). Consequently, community participation and commitment for community development—for example, identifying a community’s issues and problems, designing, and practicing decisions and policies—can be encouraged [46]. In terms of the analysis level, Coleman’s ideas of social capital facilitate the picture of a sustainable community where the allocation of such capital can uplift the community as a whole from the contribution of its members. In this “bridging” form, “the economic, social, and political relations of the community are characterized by trust that is reciprocal and diffused among members of the society” [39]. Thus, through the formal associations between members of a community as they present their affiliation in diverse social and vocational groups, “values of solidarity and norms” are integrated, extracting trust from different social interactions that cut across multi-disciplinary issues [39].

Finally, Robert D. Putnam, in a way similar to Coleman, treated social capital as a public good. He also viewed social capital as an individual feature that becomes a collective trait functioning at the aggregate levels of communities, cities, states, and nations [25,47]. Putnam et al. [25] centralized “networks, norms and trust that facilitate action and cooperation for mutual benefit” (p. 35) as features of social organizations. Therefore, his viewpoint of social capital as the amount of trust that characterizes modern political culture is a potential catalyst to reconcile Bourdieu and Coleman’s ideas of social capital, which also plays an essential part in practicing sustainability in a wider, more comprehensive way. At this ultimate level, for which “bonding” and “bridging” social capital are effective and efficient, social capital in a “linking” form is essential for development. It refers to “the structural links with decision-making institutions that are recognized as important interlocutors and toward which the engaged networks address their demands to produce development results from innovative and responsive policy decisions” [39].

The term “social capital” was coined using the analogy of capital to comprehend the role of social institutions and processes with an understanding of capital as natural resources and amenities among environmental economists [13]. After having reviewed the literature, we can see that the concept of social capital holds enormous potential to facilitate a sense of social sustainability and intensify the importance of humanity and the advantages of cooperation and collective action. Reframing social problems can, as a result, be pushed to reveal the problems beyond just the profit motive of human beings within a capitalist society to focus on the salience of human well-beings and the true meaning of an engaged and empowered society. Table 2 presents the impact that each level of social capital has for human beings and society.

Along with “risk” as a fundamental idea, we argue that social capital, as a core resource of social sustainability, can facilitate a society with vision, networks, and tools to deal with both human-made and non-human risks.

When managing and allocating social capital to deal with risks, a sustainable society should consider five key dimensions, including safety and security, equity, adaptability, social cohesion and inclusion, and quality of life. Thus, we propose a conceptual model of social sustainability consisting of these five social dimensions, interrelating with two key constitutive elements: social capital and risk.

### 3.2. Five Key Dimensions of Social Sustainability

#### 3.2.1. Safety and Security

The first dimension of a sustainable society considers safety and security. The right to be protected, to be secure, and to feel safe are fundamental parts of safety and security [2,48,49]. Regardless of such demographic and socioeconomic characteristics as race, gender, and age, a sustainable society should be a place where human beings are protected from and not exposed to vulnerable situations that can cause them harm or undermine their possibilities to avoid physical, mental, and emotional injuries and illnesses [2,50]. These situations can derive from experiencing environmental vulnerability (e.g., natural disasters and dangerous working conditions) as well as social risks (e.g., crime, violence, and riots). In order to respond well to these risks, the management of social capital plays an essential role in securing a “reliable and sufficient social security system” that ensures everyone has access to resources of basic needs such as water, food, shelter, and rest [2,14,22,23,29,51,52]. In particular, living with the unsatisfaction of basic needs are claimed to highly correlate with crime, fueling the likelihood of committing crime and increasing risks for a society. In 2008, the Basic Income Grant (BIG) Coalition implemented the world-wide first BIG pilot project in Otjivero—Omitara, Namibia, which contributed to a significant decrease in local crime. Thus, overall crime rates, according to local police’s report, fell by 42% with 43% decrease in stock theft and 20% decrease in other theft [53]. In a sustainable society, the basic constituents of human well-being are secured, and people are provided with freedom and the capability to achieve a decent level of safety [54]. For social capital, its linking form plays an important role in maintaining safe and secured environs and controlling risks, such as crime and drugs. Particularly on an institutional level in which cross-territorial-border crime might threaten human beings’ sense of security, trust, and associational partnership, as key elements of linking social capital, play an essential role in developing strategies to control crime and illegality [39]. In a best-case scenario, when social capital—especially bridging and linking social capital—is professionally managed and allocated to meet the basic needs of human beings, safety and security are achieved. In particular, in a community or neighborhood where constant efforts to minimize crime or disorder take place, residents can feel safe and secure to live in, interact with, and participate in every aspect of social life [28]. Furthermore, in a context of natural disaster, three forms of social capital could timely and appropriately assist the affected communities to cope with and recover from disaster-related damages. Bonding social networks, especially geographically regional networks, can provide food, water, and shelter if needed as well as ensure the safety and security for affected people in case of evacuation [55,56]. Bridging and linking networks are important resources for evacuation, first-aid reaction, post-trauma recovery especially basic essentials to control socially disordered risks [57] Some sample measurements were suggested to evaluate the level of safety and security:For measuring risks: crime rates, and violence rates, including domestic violence, sexual violence, youth and dating violence, child abuse, elder abuse, and technology-assisted abuse [58].For measuring basic constituents of human objective well-being: health (self-report health and life expectancy) [59].For measuring perception of safety: feeling walking alone after dark, feeling safe from serious problems of crime, feeling safe from disturbance by children/youth or traffic, and feeling comfortable/safe waiting for public transport [58].

#### 3.2.2. Equity (Justice)

“Inequality is at the root of unsustainable behaviors” [54] that enable overconsumption to be typed as social status [60], as well as disproportionately distribute resources and favor particular social groups [61,62]. In a hierarchical class structure, those with less power are likely to be affected the most from unsustainable behaviors. Fuchs [14] indicated two main reasons for this: (1) “wealth and abundance on one side and poverty and lack on the other side are an expression of a fundamental social mismatch society” and (2) “those controlling significant amounts of money, influence, reputation and social relations can more easily escape unsustainable living conditions by changing their places, contexts and forms of work and life in the case of risks and crises”. In particular, the impact of climate change has been claimed to be “socially differentiated”, resulting in different levels of vulnerability to climate-related risks among social groups [2,63]. During tender times facing with natural disasters, for example, unequally distribution of resources, external aids, and post-disaster treatments evidently exist among income groups, racial groups, gender, and age groups [63]. Consequently, climate change and natural disasters, as a type of spatial external risks, intensify social inequality including unequally distributed resilience resources and wealth inequality [64,65] Thus, equity has been one of the most essential and traditional themes in the conversation of social sustainability, including three key discussions of accessibility, equity policy, and intergenerational equity [1,18,24,66,67].

Accessibility is commonly cited as one of the foundational and effective measurements of social equity [28,48,68]. It refers to both the accessibility to the built environment and intangible opportunities through the possession of and access to social capital. A sustainable society should aim to provide equal access to the respective societies’ resources [29], which can be such essential physical settings as housing, public services, and social infrastructure. Dempsey et al. [28] indicated that there are direct and indirect accesses between some aspects of life and the built environment in which direct access is achieved. Direct access can be found either through “the actual provision of services and facilities” or “by the means of accessing them (e.g., public transport)”. Indirect access refers to the social infrastructures that should be measured by the physical quality of an infrastructure itself, as well as the services and operation of the infrastructure provided by the relevant management entities and local authorities. Other than accessibility to the built environment, accessibility to intangible resources and opportunities is pivotal to consider and move a society toward sustainability. Omann and Spangenberg [29] emphasized the notion of accessibility as the precondition for a society to be sustainable in terms of legal access, economic access, educational access, and participatory access. Thus, social capital, in its three main forms, potentially contributes to the increasing accessibility to resources, information, and support, which aligns with equity norms of sustainable development [39].

Another aspect of social sustainability requiring equity is policy. Every state, nation, and region have vulnerable groups that need special attention. Not only do vulnerable, marginalized, and disadvantaged groups have to bear a disproportionate share of environmental, social, and economic burdens, but they also are likely to be less recognized and less heard in terms of policies that hugely affect and determine their lives [69,70,71]. Significantly, social sustainability in terms of equity policy, advocates for “politics of recognition”, allowing authorities and policymakers to renavigate policies that hinder the rights of one or many social groups as well as to “deconstruct tendencies, such as queer politics, critical ‘race’ politics, and deconstructive feminism” [2]. Therefore, equity policies of a sustainable society should be aware of as well as constantly evaluate the principles, guidelines, outcomes, and effects of social, environmental, and economic policy on different social groups [2,72,73].

Adopting the notion of sustainability concerning the generational aspect of basic needs, previous research shows that equity between generations is an essential criterion of social sustainability. Specifically, intergenerational equity refers to the fairness in distribution of burdens as well as resource allocation between current generations and future generations [2,22,29]. Although starting with the intention not to compromise “the ability of the future generations to meet their needs” [8], the notion of equity has not been immersed into the generational spectrum, considering the equitable resource distribution of a society. However, the controversy of what is the measurement of intergenerational equity and how to quantify the generational level of basic needs makes this criterion seem ambiguous and abstract [50,74]. Concerning these constraints in terms of social sustainability, Eizenberg and Jabareen [2] suggested adopting the formulation of Repetto and Repetto [75], that social capital as the key resource of a society “should be managed so that we live off the dividend of our resources, maintaining and improving the asset base so that the generations that follow will be able to live equally well and better” (p. 10). Social capital, as Grootaert [76] argued, is potential to be accumulated and segmented along the spatial and racial lines, which might result in increasing inequality, even generational inequality. Therefore, allocating social capital, especially social networks, public goods and services, and integrated public support programs, to support marginalized groups is pivotal to avoid partial accumulation and move toward an even distribution of social resources and support. In other words, the widespread of social capital and concentration on needy groups would be more likely to contribute to an equitable society.

Having several areas of application, the equity dimension of a sustainable society should (1) consider the redistribution of resources of the society, especially social capital, which is the key bridging element for people to gain access to and utilize resources without overexploiting such resources; (2) constantly review and recognize vulnerable groups and address their needs to mitigate risks by allocating relevant social resources and (3) be aware of the balanced and adequate resources between the present generations and future generations to ensure the continuity of the society. To assess the equity dimension of social sustainability, some sample measurements can be applied:“Progressive taxation;Redistribution land and wealth;Reduction of unnecessary consumption in the developed world through consumption taxes on non-essentials;A public relations program highlighting the social and individual benefits of delinking materialism with social status and instead promoting sustainable behavior with social status;The return of control over economic and natural resources to local nations and communities in the developing world through nationalizing resources and industries.” [54]the accessibility of various groups of the community’s natural resources that they live in [35];the accessibility of everyday services and facilities such as doctors, food shops, newsagents, open spaces, post offices, primary schools, pubs, supermarkets, and secondary schools [22,28,77].

#### 3.2.3. Adaptability

The third dimension to consider is adaptability. While the safety and security dimension highlights the rights of human beings to be protected and secured from situations of vulnerability, adaptability refers to the competency and learning ability of a society to stay sustainable, especially during crises. Gates and Lee [49] in the Policy Report of Social Development to Vancouver City Council proposed four key guidelines of social sustainability, including adaptability. Specifically, adaptability was defined as “resiliency for both individuals and communities and the ability to respond appropriately and creatively to change. Adaptability is a process of building upon what already exists and learning from and building upon experiences from both within and outside the community” [49]. Among the sample literature collected for this study, this is one of the first documents coining this term as a social sustainability guideline, while other literature mostly expressed this aspect by tying it to the learning and progressing capacity and competency of individuals and a given society as a whole [19,22,29,35,50]. In particular, Magis [35], by assessing resilience as an indicator of social sustainability at the community level, emphasized this communal ability as the allocation and development of community resources by its members to overcome and thrive in a constantly changing environment with a high degree of unpredictability and uncertainty.

One quality of adaptability is the capacity for learning and self-organization. Learning is often referred to as a natural, individual characteristic, as individuals can grow from experiences and self-reflection; however, this is believed not to be the case for organizational learning. Learning at an organizational level requires a commitment from each and every entity to learn and organize together [50]. This practice should be performed collectively and systematically so that a given society/community can identify its strengths and weaknesses for further improvement [22]. Therefore, a sustainable society should be a place where there are no structural obstacles or systematic hindrances that limit people to transfer their learning and self-organizing experiences to an organizational level [50]. Thus, community institutions have been claimed to have the ability to persist through crises (e.g., resources depletion, social conflicts, forest fires, and border closing) and learn from challenges. As a result, to move towards a socio-ecologically resilient society, social learning and adaptive responses are essential to increase livelihood options and flexibility at multiple levels including household, community, regional, and national [78,79,80]. Conversely, in both individual and institutional levels, inadequate capacity to learn, change and improve can result in incompatibility to adapt with external changes and risks. The lack of compatible skills, knowledge, and experiences is itself a manufactured risk, which can increase vulnerability.

Additionally, adaptability also considers the utilization and application of innovations to move toward social sustainability. Innovations are not limited to only technology but can also be organizational, social, and institutional (e.g., [19,29]). Thus, the application of technical innovations can only be considered to be successful if the society accepts, welcomes, and recognizes its benefits for making a society progressive. This is one aspect of social innovation. Vallance et al. [19] discussed an example of a thin borderline between different applications in sustainable daily lifestyles that “residents may be happy to install solar panels, double glazed windows and water recycling systems but may draw the line at ‘transformative’ composting toilets (which involve a more intimate engagement with human waste than standard ‘flush it away’ models), or moving from suburban settings to high-density, apartment-style living arrangements”. Trivellato [81] showed that by integrating the notion of social sustainability into the Smart City Strategy, governments and authorities of Milan, Italy, through innovations, have encouraged more flexibility in their urban planning strategy, including bottom-up projects and a high degree of responsiveness and opportunities for different actors of the city. Therefore, a sustainable society should support innovations through “the extension of societal and company participation, education, and higher expenditures for research and education” [29], which allow more flexibility, systematic learning, and self-organizing capacity to deal with both inner and outer risks, especially uncontrollable environmental threats. In a sustainable society, social capital, in bonding form, is useful to boost adaptability, especially for low-income and social excluded groups, by sharing knowledge, information, and financial risk as well as claiming for “reciprocity in times of crisis” [82]. With bridging social capital is argued to be important particularly under the circumstances of dynamic mobile communities and managing collective resources [35,82]

Measuring resilience levels can be accomplished through sample metrics such as these:Subjective resilience: perception and belief of individuals in their ability to affect the community or society’s well-being [35]; perception of individuals’ preparedness and readiness to respond to risks.Objective resilience: “The effectiveness of community government in dealing with important problems facing the community;” “the extent to which communities/societies affected by change attempt to keep things the same or try new ways of doing things;” “changes in the community’s capacity over time to respond to change, develop a new future for itself, and develop and implement community-centered plans” [35].

#### 3.2.4. Social Inclusion and Cohesion

In addition to ensuring the notion of equity by recognizing and uplifting disadvantaged and marginalized groups, a sustainable society should consider a balance that creates social inclusion and cohesion by encouraging public participation and engagement [39,83] and enhancing social trust from every social group within [50]. Bramley and Power [84] argued that social sustainability, if being treated as social development, is likely to be equated to social capital, social cohesion, and social exclusion. From this viewpoint, social exclusion is one of the risks that a sustainable society needs to fight. To tackle the risk of a social group being excluded or constrained with judgments and prejudices, enhancing social trust as the highest form of social capital is a must [50].

First, to boost social trust, a sustainable society needs to increase its members’ constructive contribution to public dialogue in a collective manner that fosters local commitment and belief in the legitimacy of the society/community’s needs, as well as the belief in the personal values of each member [20,28,85]. For example, Natcher and Hickey [86], in their work of examining community-based resources, suggested the optimistic side of involving indigenous people in the resource management process as a criterion of social sustainability. Thus, including community members who belong to different social groups, or at least their representatives, can empower those individuals and provide them with recognition of their values [86,87]. As a result, collective decision-making process and effective bottom-up strategies would potentially be the policy applications achieving from strong social networks between community members, authorities, and governments, especially in the context of climate-related disasters [88]. This idea aligns with the notion of a sustainable community where diversity is celebrated [22,49,50] in terms of ethnical, cultural, and socioeconomic background [89].

Second, a strong sense of community can be a catalyst to improve the social inclusion and cohesion even further [28,50,85]. What needs to be clarified here is that a sense of community should not be associated synonymously with a sense of place, but instead encompass a broader understanding of both attachment to a physical environment and a given community’s collective values. Transforming a sense of community to personal motivation for a collective contribution to community should be the pathway worth pursuing, as missing this factor can lead to some forms of dependence and inflexibility in dealing with risks, such as natural disasters [90,91,92]. Thus, other studies have shown that social fragmentation and lack of social cohesion is very likely to hinder a society from building its socio-ecological resilience [93] Particularly, in a study of the relationship between the idea of a sustainable society and the idea of a social-ecologically resilient society, Baldwin and King [94] exemplified big cities, from the perspectives of migrants, as a “workplace” where migrants “hardly imagine contributing to the climate change and health resilience”.

Last, a strong and sustainable community/society should promote the connectedness by providing processes, systems and structures not only within, but also outside the community at informal, formal, and institutional levels [22,23]. This element is especially essential for a community/society, as they can turn to and seek additional assistance from the outside during crises. Walker et al. [95], in a study of social-ecological resilience, adaptability, and transformability, claimed that the social components of a socio-ecological system consist of diverse groups of people in different levels and with different viewpoints, contributing to the whole panarchy of a society and affecting the latitude, resistance level and precariousness of a society’s resiliency. Therefore, these diverse sub-systems need to be included, heard, and connected to provide a thorough understanding of what is desirable and undesirable for all the members of the society.

We suggest sample metrics that have been proposed by the previous literature:the level of public participation in communal activities, and collective networks in a community [28,35];the level of integrating and including various groups among community institutional organizations; the extent to which community decision-making processes engage diverse perspectives and reflect cultural differences [35];“The extent to which people from diverse groups share support, resources, knowledge, and expertise when confronted with change;” [35]“The extent to which community members look outside the community to find resources to support their endeavors” [35].

#### 3.2.5. Quality of Life

In today’s capitalist society, “welfare economics, in its current form, has been very successful in enhancing material well-being, but not for everybody” [37]. Social well-being is likely to restrict to economic indicators that economic abundance is somehow assumed to be correlated directly with elevated levels of happiness. However, Jackson [96] argued that there are essential social indicators that better measure the social well-being of a society such as family, health, education, and social relationships. Even though happiness has been discussed among philosophers for thousands of years, its actual measurement has only been focused on as an alternative approach to assess human beings’ overall happiness index within the past few decades [37]. Recognizing the unnecessary direct correlation between economic success and happiness, we propose that the last element of social sustainability is “quality of life”. This dimension addresses higher levels of human beings’ well-being excluding the basic need to survive or, from another perspective, subjective well-being. Measuring the subjective aspect of social well-being, as Rogers et al. [54] suggested, offers an insight of social and emotional state of individuals that somehow varies across social groups and social circumstances. Embracing both objective and subjective sides can construct a comprehensive understanding of social well-being which are highly likely to vary and depend on individuals’ perceptions, experiences, and desires. A sustainable society, in this sense, should be a place for every member to choose how they want to live and work [24,29,51]. Democracy and autonomy should be promoted so that people can achieve self-actualization through different pathways such as education, recreation/leisure, social relationships, and social fulfillment [23]. Thus, social prejudices and oppression could perform as manufactured risks that seriously affect some particular groups. For example, 2020 annual National Preparedness Report from the US Federal Emergency Management Agency (FEMA) shown that LGBTQ people are more likely to suffer from social isolation, disrespect, and harassment in settings as emergency shelters [97]. The risk of social prejudices and oppression, in this case scenario, does not originate from biological features but from how a social group choose to live their lives. Therefore, social capital, in this dimension of increasing quality of life, holds significant potential to assist socially prejudiced and oppressed groups to move toward self-actualization and true happiness. In particular, based on the analysis of social capital means, output, and outcomes in Table 2, bonding social capital, through self-help and help from informally social circle, can help creating a safe place, physically and mentally. Bridging social capital expands more to formal networks that people can search for help with sectoral programs advocating for human well-being and happiness such as programs supporting LGBTQ’s rights. Linking social capital, presenting in forms of interconnected networks between decision-making institutions, can result in faster and more integrated policy responses against social prejudices and oppression [39]. Clark [98] measured the psychology of human well-being as being constituted from “mental functioning, pleasure, joy, avoiding stress and frustration, self-confidence and status” [54] as well as capturing some “better things” for better lives, including free time and recreation, leisure, being with family and friends, and religion and church.

To measure this dimension, a comprehensive happiness index should be constructed to capture the subjective aspects of life, including: “identity, autonomy, and self-determination; freedom to move about and choose a job; home and social relationships; education and knowledge; fulfillment and creative outlets; and time and space for recreation, connection with nature and beauty, and hope for the future” [54]. Some leading measuring tools that have been developed are:the Satisfaction with Life Scale (SWLS) [99];the Cantril Ladder Method, which was used in the World Happiness Report [100].

## 4. Discussion

This paper proposed a comprehensive conceptual model of social sustainability consisting of five main dimensions: (1) safety and security; (2) equity; (3) adaptability; (4) social inclusion and cohesion; and (5) quality of life. Through the prism that each dimension offers, two key constitutive components of social sustainability such as risk and social capital should be assessed. In Figure 1, we propose social capital as the core resource that holds the potential of a society to progress toward a sustainable state if it is well-managed and allocated [8,101] while risks are recognized, mitigated, and gradually eliminated [2]. It is a performative and relational structured model in which five dimensions are interrelated and supportive toward each other to reframe a vision for a more sustainable society in the future. As the world and society are constantly active and changing, social sustainability in general and our conceptual model specifically should be seen as a manner, a guideline, and a social process rather than a fixed goal to pursue [13].

One of the dominant challenges of conceptualizing social sustainability is the complexity and heterogeneity of the current social life [19,29]. This poses a burden on scholars, policymakers, and practitioners to understand this in a conceptual perspective; instead, comprehending this concept through various levels, viewpoints, and disciplines appears to be more accessible. Therefore, our conceptual model is not a “one-size-fits-all” model that can be applied in every social circumstance, context, and geographical territory. Williams et al. [15] suggested that “any change in non-crisis circumstances to a more sustainable future must be embedded in deep, already existing cultural themes” (p. 34). Thus, our conceptual model serves as a guideline for those who aim to pursue social sustainability for their local neighborhoods, communities, regions, and nations. Applying the model requires context-specific awareness and understanding of the practitioners [54] in order to achieve multidimensional, place-based, process-oriented, and culturally diverse solutions and values for a sustainable society [20,27,28,102,103]. Thus, understanding and centering locality and their cultural values should be considered as the step one before any action is taken. Accordingly, we argued for the bottom-up approach of social sustainability policy which enhances “the dimensionality of our human experience and favors diversity” as the way “humanity has created all forms of durable societal organization, including hierarchies” [39,103].

The sustainability concept has a strong and dynamic policy background with the active discussion among institutions such as the United Nations and the European Union [14]. Additionally, the sustainability concept was coined and actively discussed in the capitalist society where class and power occupy every sector of social life. As we constructed and proposed this model, the central elements are human beings. In understanding inequality as the heart of an unsustainable society, this concept should not be taken as a call to intensify structural stratification and practice power of the minority over the majority through politics [14,27]. Our conceptual model should not be adopted as an exception that social sustainability policy should be a tool to serve every member of any given society, or as an answer for one single question: “What do we need and want to sustain?”

Baldwin and King [94] argued that “socially sustainable communities can also be resilient communities”. Indeed, at a community level, the idea of social sustainability has been proposed with overlapping concepts and dimensions to the idea of socio-ecological resilience. These dimensions include humans’ well-being, safety and security, and social solidarity [28,94]. In this study, the context of natural disasters is used to discuss two constitutive elements—risk and social capital—through five key dimensions of social sustainability. Dealing with natural disasters is also a common context for every society, community, and entity to evaluate its socio-ecological resiliency. Additionally, according to ARUP [104], there are four dimensions of cities’ resilience, including health and well-being, economy and society, infrastructure and environment, and leadership and strategy. These dimensions consider human beings as the center which aligns closely with five key dimensions of a socially sustainable society that we proposed. Therefore, we claim that a socially sustainable society is congruent with a socio-ecologically resilient society in providing its citizens with safety and security, equity, adaptability, inclusivity and coherence, and autonomy to achieve their desired quality of life. As Baldwin and King [94] emphasized in their study, “social sustainability is important because without strong networked, cohesive communities, the human capital required to build, run, and maintain sustainable, resilient cities will wither.” (p. 138).

## 5. Conclusions

Modern society is a risk society [31,33,34], in which human beings have been increasingly dealing with threats, especially severe external threats such as climate-related disasters. However, the management and allocation of social capital as key resources and assets can contribute to moving toward a constantly sustainable society. For a society to be sustainable, security, equity, and social inclusion are not the only elements to consider, but adaption and adaptability to external changes of the environments of the society are also vital to pursue. Inversely, the sustainable state of a society would also reinforce the resilience level of the society pre-, during, and post-handling of natural disasters and climate change. To do so, allocating social capital is vital to help people in need to (1) stay safe and secured, (2) equitably access to resources, (3) be aware, learn, and adapt, (4) be heard and involved, and (5) be happy. A socially sustainable society can also be a resilient one, especially against climate-related risks and disasters and vice versa [35,94].

Therefore, further research of social sustainability is encouraged, and it requires multi-disciplinary efforts. The model and theoretical suppositions should be assessed and improved with empirical data and examples from practice for the sake of validity and reliability.

## Figures and Tables

**Figure 1 ijerph-20-05350-f001:**
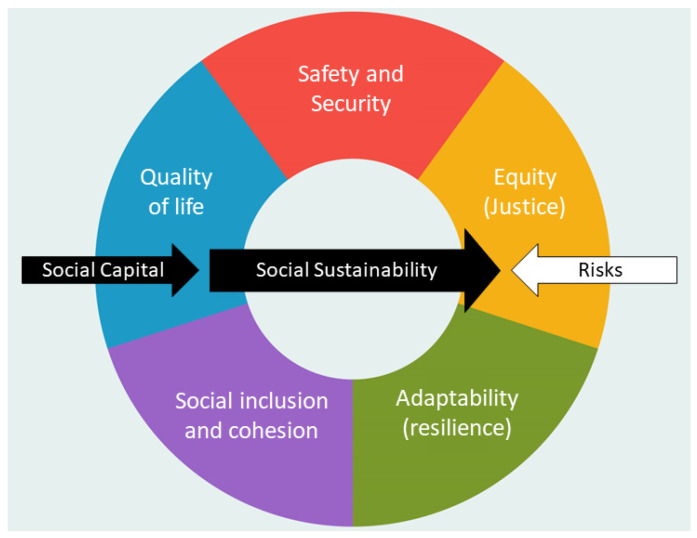
Conceptual model of social sustainability.

**Table 1 ijerph-20-05350-t001:** Representative definitions of social sustainability.

Authors/Year	Discipline	Key Idea
Stren and Polese [18]	Urban planning	“Social sustainability for a city is defined as development (and/or growth) that is compatible with the harmonious evolution of civil society, fostering an environment conducive to the compatible cohabitation of culturally and socially diverse groups while at the same time encouraging social integration, with improvements in the quality of life for all segments of the population”
Harris et al. [21]	Economics	“A socially sustainable system must achieve fairness in distribution and opportunity, adequate provision of social services, including health and education, gender equity, and political accountability and participation”
McKenzie [22]	Social Sciences	“Social sustainability is a positive condition within communities, and a process within communities that can achieve that condition”
Littig and Griessler [23]	Sociology	“Social sustainability is a quality of societies. It signifies the nature-society relationships, mediated by work, as well as relationships within the society. Social sustainability is given if work within a society and the related institutional arrangements satisfy an extended set of human needs are shaped in a way that nature and its reproductive capabilities are preserved over a long period of time and the normative claims of social justice, human dignity and participation are fulfilled”
Magis and Shinn [24]	Public Administration	“Social sustainable concerns the ability of human beings of every generation to not merely survive, but to thrive”
Vallance et al. [19]	Urban Planning	Threefold schema of social sustainability comprises: “Development sustainability addressing basic needs, the creation of social capital, justice, and so on; Bridge sustainability concerning changes in behavior so as to achieve bio-physical environmental goals; Maintenance sustainability referring to the preservation—or what can be sustained—of socio-cultural characteristics in the face of change, and the ways in which people actively embrace or resist those changes”
Eizenberg and Jabareen [2]	Urban Planning	“Social sustainability is constituted with ‘risk as the ontological foundation of sustainability’ and equity, safety, sustainable urban forms, and eco-prosumption as four main components”

**Table 2 ijerph-20-05350-t002:** Results produced by social capital: Means, outputs, and outcomes (adapted with permission from Ref. [39]. 2016, Raffaella Y. Nanetti and Catalina Holguin).

Forms	Means	Output	Outcome
Bonding	Self-help/informal social circle	Ad hoc services/assistance to groups	Increased well-being of groups
Bridging	Formal associational networks	Sectoral programs and actions	Sectoral development
Linking	Coordinated policy demands	Integrated development policies	Sustainable territorial development

## Data Availability

Not applicable.

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
