# Peer review of "New Conceptual Model of Social Sustainability: Review from Past Concepts and Ideas"

_ijerph, 2023, doi:10.3390/ijerph20075350_

Round 1

Reviewer 1 Report

Thank you for the opportunity to review the manuscript entitled "New Conceptual Model of Social Sustainability: Review from past concepts and ideas"

As a specialist in sustainability transitions, I was anticipating a thoughtful contribution to the developmental discourses on this theme. However, having read the manuscript a couple of times now, I remain disappointed and frustrated with the submission. In the following paragraphs I outline the origins for this response.

The first impression upon reading the manuscript is that the level of English used almost throughout the paper requires extensive revision. It is not at the required standard of academic discourse, and actually detracts from the reader's engagement with the content of the paper. There are multiple examples of poor syntax, grammar and missing words peppered throughout the manuscript, so on this basis alone, the paper requires extensive revisions.

This being said, the content of the paper is potentially interesting, but is lacking in some key aspects. These are discussed below.

It is difficult to fathom what the main point(s) of the paper is (are). The authors use the paper to develop an argument that academics, researchers, and policy-makers have tended to avoid articulating a model of a sustainable society, and then go on to propose a five point model that they claim will help inform academics, researchers, and policy-makers in developing an evaluation framework for a resilient and adaptive society.

However, the initial premise of the paper is itself flimsy and doesn't bear up to even superficial scrutiny. For example, following the claims of a few articles, the authors of this paper argue that social sustainability is the missing pillar in the sustainable development discourse. The authors note the contribution of the Brundtland Commission, which clearly discusses the role of social sustainability, and the emphasis on sustainable economic models has predominated much of the sustainability discourses since the late 1980s. I'd warrant that most economists would argue that their work is for the good of society and its resilience, even though of course, ecological economists would challenge this.

The point being, the matter of what constitutes a sustainable society has long been a matter of robust discussion and debate, although this is summarily dismissed as 'chaos' by the authors (see below). It appears that the authors have not familiarised themselves with the work over the last decade or more in the disciplines of environmental sociology and environmental humanities. On a more trivial note, reference to academic 'chaos' regarding the notion of sustainable societies suggests an internal contradiction: if the matter has not been debated, etc. as the authors imply, then how does one encounter a situation of academic chaos?

A second challenge to the author's premise is that of the 17 SDGs, easily 10 or more are directly human (society) oriented, so the argument that social sustainability is a missing piece of the larger jigsaw doesn't really track, even at this superficial level of analysis. Indeed, the authors already acknowledge this when they write (lines 144 - 146) "While the discussion of sustainability with a specific focus on diverse expressions of social sustainability has been [sic] flourished among practitioners, the academia [sic] seems to [sic] less involved in the ideology of social sustainability". The reader is not offered any insight into what this 'ideology of social sustainability' might be, but the authors then go on to claim that (lines 150 - 152) "The idea has been scattered studied [sic] and understood diversly [sic], creating an academic chaos and adding to the vagueness of the social dimension". Quite what this means went over my head.

In a later section, the authors introduce the concepts of risk and resilience as these pertain to social sustainability, and argue that a sustainable society is - in effect - one that can adapt to and survive disasters. If this is what the authors mean by their reference to the missing 'ideology' (lines 144 - 146), then again this demonstrates a limited knowledge of the available literature. They need only dip into the work undertaken by, inter alia, the Resilience Alliance, or quickly review individual articles and special issues of the open access and highly respected journal 'Ecology & Society' to see ample evidence that social resilience and responses to disasters, both rapid and slow-burning, are indeed at the forefront of many contributors' research efforts. This is to say, there is already an ample and steadily growing literature in this domain, but the authors fail to differentiate what constitutes 'chaos' from what constitutes the 'ideology', and still do not clearly state what it is that they are offering that is either different or that 'cuts through' the 'chaos'.

In sum, the authors' core premise just doesn't track with abundant evidence to the contrary and smacks of a straw-man argument that is used simply to be knocked down to shoe horn their five point model into the debate. What the model is meant to contribute to this debate is not declared.

So, what then of their five point model which is the core contribution the authors offer? These five "social dimensions" are to be viewed via the dual prism of 'social capital' (effectively, social relationships and bonds among people) and 'risk'. These five dimensions are: 'safety & security', 'equity', 'adaptability (resilience)', 'social inclusion and cohesion', and 'quality of life'.

Each is reasonably well referenced, although the examples used to illustrate these have tenuous links with what the authors purport to describe. For example, for the first of the five dimensions, the authors use crime rates, health rates, public perceptions of safety as examples of 'safety & security', although how these relate to social capital and the nature of risk (as elaborated in their section 3.1.) is not made clear. Are they suggesting that, with reference to the literature on risk, for example, that modernisation is at the heart of perceptions of public safety or human health? It is difficult to be sure, because the section on this dimension idealises communities and neighbourhoods "free from any crime and disorder" (line 375). Perhaps increased funding in police or community safety officers would be a way to address this, but the role of social capital per se in this arrangement remains undeveloped. The alleged contribution of this to promoting social sustainability is, again, implied rather than elaborated, even though the onus is on the authors to make their case.

This is not to dismiss the entirety of the authors' contributions however. There are some valid arguments, and ideally one can but aspire to live in an equitable worlds, a world within which there are strong social bonds, and so on.

However, at its present stage of development, this manuscript is below standards one would expect of this journal. In sum: the paper is poorly written, to the point of being almost unintelligible in parts; the arguments are selective and seemingly cherry picked in order to generate a point of departure for their contribution, which is an as yet under-developed conceptual model of a sustainable society.

As such, this paper runs the real risk of adding to the academic chaos it claims other works have resulted in, rather than the clarification and specificity the authors' propose they are seeking. For this reason, while I am tempted to recommend that this paper be rejected, I would like to encourage the authors to take a second go at this, to address these concerns and criticisms substantially, and to resubmit a manuscript that is of a much improved quality than this version.

Author Response

We thank the academic editor, and four anonymous reviewers for the opportunity to revise and resubmit our manuscript to International Journal of Environmental Research and Public Health. The reviewers provided us with thoughtful and thorough evaluations of our paper. We believe that the revisions we have made based on these comments have allowed us to develop a much-improved manuscript. However, if we have missed something, we are more than willing to make the necessary changes going forward. Our responses to the specific points raised by each editor and reviewer are outlined below. Please find our responses in the attachment.

Reviewer 2 Report

The article, “New Conceptual Model of Social Sustainability…” seeks to fulfill a valid task of mapping out and analyzing various framings of social sustainability in the scholarship. However, in my view, the article does a rather limited job in accomplishing such goal.

The article presents several important idea and concepts in the findings section, but it fails to explain how these themes and ideas are derived from a systematic review of the social sustainability scholarship.

The authors need to actually explain how they sampled articles for review, how they analyzed these article, coded the data, and arrived to the themes reported in the findings section.

Author Response

(The authors gave the same response as above.)

Reviewer 3 Report

Very interesting subject, very good use of the English language, sound bibliographic research on the notion of social sustainability. Concerning resilience, though, there was use of terms and definitions in the article, with a different context than the context proposed in other research. The fact that these differences are not mentioned/analyzed in the article, might be a cause of confusion.

According to the authors, social sustainability has five key dimensions: safety and security, equity (justice), adaptability (resilience), social inclusion and cohesion, and quality of life. Adaptability (resilience) in turn has as important qualities the communal ability for allocation and development of resources in a community, the capacity for learning and self organization, and the utilization and application of innovations.

According to B. Walker, C.S. Holling, S.R. Carpenter and A. Kinzig ("Resilience, adaptability and transformability in social-ecological systems" Ance Ecology & Society, vol 9, no 2, art 5, JSTOR, 2004) there are three attributes of social-ecological systems: resilience, adaptability, and transformability. Resilience in turn, has four components: latitude, resistance, precariousness and panarchy. Adaptability is the capacity of actors to influence resilience by altering the components of resilience, and transformability is the capacity to create a fundamentally new system from an existing one. Here, resilience is considered as a different attribute than adaptability, and there is a different approach for defining and classifying resilience.

Finally, in the resilience approach of ARUP (City Resilience Index, 100 Resilient Cities Programme) there are four dimensions of resilience (Leadership and strategy, Health and wellbeing, Economy and society, and Infrastructure and ecosystems).

Here we see that four out of five key dimensions of social sustainability (with the exception of resilience) in the present article, are included in the four dimensions of resilience in the ARUP approach. The impression that a reader might get by considering the article on the one hand, and bibliography on resilience on the other, is that there is a tendency in the article to reduce the notion of resilience to one of the five dimensions of social sustainability, by omitting some of its properties and characteristics that are recognized as such in resilience bibliography.  

Author Response

(The authors gave the same response as above.)

Reviewer 4 Report

Concepts of social sustainability review

·       Interesting paper

·       References needed in terms of debates in the introduction

·       Signposting the structure of the paper would be useful

·       Why is the discussion of social sustainability important in the context of disasters (and why climate related disasters?)

·       Why is history and policy movement important to the debate the paper wants to focus?

·       Summary table of different definitions in discipines interesting

·       Why were these five dimensions considered key? Are there any other dimensions that could be important as well?

·       Adaptability is not really resilience. Resilience is more than adaptability

·       Very interesting paper trying to synethise the literature on social sustainability and create a framework. More justifications on why these five factors are considered vital to the framework would be useful. How the framework stitch together would be useful too? What about the temporal element? What is the role of culture? 

Author Response

(The authors gave the same response as above.)

Round 2

Reviewer 1 Report

The authors are to be commended for their efforts to improve the original manuscript and adopt the changes requested. This has resulted in a more coherent and cogent paper.

There are a few minor typos and grammatical issues that will require a final thorough proof-reading prior to submission, but once this has been done, I am happy to support publication.

Author Response

We thank you for your comments on our manuscript to International Journal of Environmental Research and Public Health. You provided us with thoughtful and thorough evaluations of our paper. We believe that the revisions we have made based on these comments have allowed us to develop a much-improved manuscript. However, if we have missed something, we are more than willing to make the necessary changes going forward. Our responses to the specific points are outlined below. Please find our responses in the attachment.

Reviewer 3 Report

The relationship between social sustainability and socioecological resilience still unclear and not adequately researched.

Author Response

(The authors gave the same response as above.)

Reviewer 4 Report

The paper has addressed most of my concerns and it is more robust now. 

Author Response

(The authors gave the same response as above.)
